# MARCH8 Suppresses Tumor Metastasis and Mediates Degradation of STAT3 and CD44 in Breast Cancer Cells

**DOI:** 10.3390/cancers13112550

**Published:** 2021-05-22

**Authors:** Wenjing Chen, Dhwani Patel, Yuzhi Jia, Zihao Yu, Xia Liu, Hengliang Shi, Huiping Liu

**Affiliations:** 1Department of Pharmacology, Feinberg School of Medicine, Northwestern University, Chicago, IL 60611, USA; wchen@mcw.edu (W.C.); dbp5384@psu.edu (D.P.); y-jia@northwestern.edu (Y.J.); ZihaoYu2021@u.northwestern.edu (Z.Y.); Xia.Liu@uky.edu (X.L.); 2Department of Toxicology and Cancer Biology, University of Kentucky, Lexington, KY 40506, USA; 3Institute of Digestive Diseases, Xuzhou Medical University, Xuzhou 221006, China; 4Central Laboratory, The Affiliated Hospital of Xuzhou Medical University, Xuzhou 221006, China; 5Department of Medicine, the Division of Hematology and Oncology, Feinberg School of Medicine, Northwestern University, Chicago, IL 60611, USA; 6Lurie Comprehensive Cancer Center, Northwestern University Feinberg School of Medicine, Chicago, IL 60611, USA

**Keywords:** MARCH8, breast cancer metastasis, CD44, STAT3

## Abstract

**Simple Summary:**

Protein ubiquitination is catalyzed by many enzymes, whose functions and substrate specificity are not fully understood. This study reports the expression patterns of the membrane-associated RING-CH (MARCH) family members in breast cancer and their association with patient outcomes. Specifically, MARCH8 is a newly identified tumor suppressor with a role in inhibiting breast cancer metastasis and enhancing cancer cell death. MARCH8 not only promotes the degradation of membrane proteins such as the breast cancer stem-cell marker CD44 through the lysosomal degradation pathway, but also recruits a previously unknown nonmembrane target protein, signal transducer and transcription activator 3 (STAT3), for proteosome-dependent degradation.

**Abstract:**

Protein stability is largely regulated by post-translational modifications, such as ubiquitination, which is mediated by ubiquitin-activating enzyme E1, ubiquitin-conjugating enzyme E2, and ubiquitin ligase E3 with substrate specificity. Membrane-associated RING-CH (MARCH) proteins represent one novel family of transmembrane E3 ligases which target glycoproteins for lysosomal destruction. While most of the MARCH family members are known to degrade membrane proteins in immune cells, their tumor-intrinsic role is largely unknown. In this study, we found that the expression of one MARCH family member, MARCH8, is specifically downregulated in breast cancer tissues and positively correlated with breast cancer survival rate according to bioinformatic analysis of The Cancer Genomic Atlas (TCGA) dataset. MARCH8 protein expression was also lower in a variety of human breast cancer cell lines in comparison to immortalized human mammary epithelial MCF-12A cells. Restoration of MARCH8 expression induced apoptosis in human breast cancer cell lines MDA-MB-231 and BT549. Stable expression of MARCH8 inhibited tumorigenesis and lung metastases of MDA-MB-231 cells in mice. Moreover, we discovered that the breast cancer stem-cell marker and metastasis driver CD44, a membrane protein, interacts with MARCH8 and is one of the glycoprotein targets subject to MARCH8-dependent lysosomal degradation. Unexpectedly, we identified a nonmembrane protein, signal transducer and transcription activator 3 (STAT3), as another essential ubiquitination target of MARCH8, whose degradation through the proteasome pathway is responsible for the proapoptotic changes mediated by MARCH8. These findings highlight a novel tumor-suppressing function of MARCH8 in targeting both membrane and nonmembrane protein targets required for the survival and metastasis of breast cancer cells.

## 1. Introduction

Protein ubiquitination (or ubiquitylation) is one of the best known post-translational modifications. It is coupled with protein localization and stability in many essential functions of mammalian cells, such as the cell cycle, oncogenic transformation, and immune cell functions [1,2]. This process is catalyzed by multiple sequential enzyme classes (termed E1, E2, and E3) to form a thiol-ester linkage between the C-terminus of ubiquitin and a lysine (Lys or K) either on a target protein or on the last ubiquitin of a polyubiquitin chain coupled with a target protein. The unveiling of a growing number of E3 ubiquitin ligase families with a broad spectrum of protein substrates has resulted in the accelerated development of protein-targeting strategies in immune diseases and cancers [1,2,3,4,5,6].

Three major categories of E3 ligases include RING (really interesting new gene), HECT (homologous to the E6AP carboxyl terminus), and RBR (RING between RING) families [7,8,9]. Belonging to the family of RING E3 ligases, the membrane-associated RING-CH (MARCH) family consists of 11 recently identified members, with a characteristic RING-CH domain distinct from the classical RING finger domain containing eight cysteine and histidine residues [9,10,11,12,13]. MARCH proteins were first identified as mammalian homologs of viral E3 ligases K3 and K5, which are involved in immune evasion [10,11,12,13]. Most of the MARCH family members have been linked to immune cell regulation by targeting membrane protein targets; however, their tumor-intrinsic roles are largely unknown. Our research program is poised to investigate the clinical relevance and molecular functions of MARCH proteins in breast cancer. In this study, we found that MARCH8 is specifically downregulated in breast cancer, especially triple-negative breast cancer, which lacks targeted therapies and frequently metastasizes to distant organs with unfavorable outcomes.

MARCH8, originally termed as cellular modulator of immune recognition (c-MIR), was the first identified human E3 ligase of the MARCH family that plays important roles in the immune response [10]. It is located on endosomal and cytoplasmic membranes, with a cytosolic N-terminal RING-CH domain and two transmembrane domains with a loop region, linked to a short cytosolic C-terminal tail [9,10,11,14]. MARCH8 is ubiquitously expressed in many human tissues and cell types. However, its role in breast tissue has yet to be elucidated. MARCH8 has been shown to downregulate a variety of cell membrane receptors in immune cells such as major histocompatibility complex I (MHC I) HLA 2.1, MHC II, CD95 (Fas), B7.2, TfR, CD166, CD88, and CD98 [9,10,11,14]. However, its tumor-intrinsic role is less understood. In this study, we discovered both membrane and nonmembrane protein targets of MARCH8, i.e., CD44 and signal transducer and transcription activator 3 (STAT3), respectively, in different subtypes of breast cancer cells. 

CD44 is a cell surface transmembrane glycoprotein enriched in breast tumor-initiating cells (or cancer stem cells) [15]. CD44 overexpression is positively correlated with invasive and metastatic breast cancer with a poor prognosis [15,16]. High CD44 levels have also been discovered as a marker for cancer stem cells in many other solid malignant tumors [17,18], modulating intracellular pathways via protein interactions and STAT3 signal transduction [19]. Our previous studies demonstrated that homophilic CD44 interactions mediate tumor stem-cell aggregation and polyclonal metastasis [15,20,21]. However, the regulatory mechanisms underlying CD44 protein stability were unclear. Our finding that MARCH8 can interact with and downregulate CD44 highlights a possible targeting strategy for breast cancer. 

STAT3 is constitutively activated by phosphorylation of tyrosine 705 (Y705) in many cancers, acting as a point of convergence for oncogenic signaling pathways and promoting tumor cell survival and metastasis [22,23,24]. It has been reported that STAT3 is subject to viral protein-dependent ubiquitination and degradation [25]. However, the molecular mechanisms underlying the regulation of STAT3 stability and degradation in the context of cancer cells are unclear. The current study highlights a previously unknown function of MARCH8 in ubiquitinating nonmembrane protein STAT3, which results in proteasomal degradation and triggers proapoptotic signals in breast cancer cells.

## 2. Materials and Methods

### 2.1. Animal Studies

All mouse maintenance and procedures were performed following the NIH Guidelines for the Care and Use of Laboratory Animals and approved by Northwestern University’s Institutional Animal Care and Use Committee (IACUC) (protocol # IS00004667). The mice used in this study were kept in specific pathogen-free facilities in the Center for Comparative Medicine at Northwestern University. The animal sample sizes were determined on the basis of statistical analysis in preliminary experiments.

Female NOD.Cg-Prkdc^scid^Il2rg^tm1Wjl^/SzJ (NSG) (Jackson Laboratory, Bar Harbor, ME USA, Cat 005557) mice, 6 to 8 weeks old, were used for human MDA-MB-231 cell-based xenograft studies. For assessing the tumorigenic potential of MDA-MB-231 cells, cells labeled with Luc2-tdTomato (L2T) were trypsinized, and 100 cells were injected orthotopically into the second and fourth mammary fat pads of NSG mice after mixing with Matrigel (1:1 ratio) (Thermo Fisher Scientific, Waltham, MA, USA, Cat 354234) as described [20]. Tumor growth was monitored by bioluminescence imaging.

### 2.2. Bioinformatic Analysis

Using the online Gene Expression Profiling Interactive Analysis (GEPIA) platform accessed on 19 March 2021 (http://gepia.cancer-pku.cn/index.html) [26], we performed boxplot analyses (expression DIY) of MARCH family gene expression in human tumors versus paired normal tissues in The Cancer Genome Atlas (TCGA) dataset, including breast cancer (BRCA) and other tumor types. Specifically, the *Y*-axis gene expression in boxplots is presented as log_2_ (transcripts per million + 1) on a log scale. Asterisks represent significant differential expression between tumor (T) and paired normal (N) tissues in TCGA, with cutoffs of log_2_ (fold change) = 0.5 and *p* < 0.05. A jitter size of 0.4 was used in the box plots. Using the Prognoscan database, we analyzed the correlation between gene expression and survival in clinical patient samples [27]. 

### 2.3. Cell Culture

Purchased from ATCC, the HEK-293, mammary epithelial MCF-12A, and human breast cancer cell lines MCF-7, BT-549, SKBR3, BT-474, and MDA-MB-231 were cultured in Dulbecco’s modified Eagle medium (DMEM), high glucose (Thermo Fisher Scientific, Waltham, MA USA, Cat SH30243FS), supplemented with 10% fetal bovine serum (FBS) (Thermo Fisher Scientific, Waltham, MA, USA, Cat 16000044) and 1% penicillin–streptomycin (Thermo Fisher Scientific, Waltham, MA, USA, Cat SV30010). All cell lines were routinely verified as mycoplasma-free as analyzed by the MycoAlert™ PLUS Mycoplasma Detection Kit (Lonza, Basel, Switzerland Cat LT07-703). For MDA-MB-231 cell suspension culture, cells were trypsinized into single-cell suspensions and transferred to poly-hydroxyethyl methacrylate (poly-HEMA) (Sigma-Aldrich, Darmstadt, Germany, Cat P3932-10G) coated plates. Before use, these six-well plates received 700 µL per well of 20 mg/mL poly-HEMA reconstituted in 95% ethanol and dried overnight in a tissue culture hood, sterilized by UV for 1 h, and washed three times with phosphate-buffered saline (PBS). Cells were suspension-cultured for 24 h and then collected for immunoblotting. 

### 2.4. Cell Transfection, Transduction, and Treatment

The MDA-MB-231 cells were lentivirally labeled by L2T and MARCH8-GFP (OriGene Technologies, Rockville, MD, USA, Cat RC209891L2) or GFP control using the lentiviruses and labeling protocol previously described [20]. Transfections of MDA-MB-231, BT-549, and HEK-293 cells were performed using Lipofectamine LTX Reagent (Thermo Fisher Scientific, Waltham, MA, USA, Cat 15338100), Lipofectamine 3000 (Thermo Fisher Scientific, Waltham, MA, USA, Cat L3000001), and FuGENE HD Transfection Reagent (Promega, Madison, WI, USA, Cat E231A), respectively, according to the supplier’s instructions. To block the proteasomal degradation pathway in MDA-MB-231 cells, a 1:1000 dilution of MG-132 (Abcam, Cambridge, UK, Cat ab147047) at a final concentration of 10 µM was used to treat the cells for 6 h to block proteosome degradation pathway. A 1:1000 dilution of chloroquine (Cell Signaling Technology, Danvers, MA, USA, Cat 14774S) at a final concentration of 50 µM was used for 24 h to block the lysosomal degradation pathway. 

### 2.5. Antibodies and Plasmids

The primary antibodies that were used in our experiments include MARCH8 (Proteintech, Rosemont, IL, USA, Cat 14119-1-AP), BAX (Cell Signaling Technology, Danvers, MA, USA, Cat 5023S), BID (Cell Signaling Technology, Danvers, MA, USA, Cat 2002), cleaved caspase-3 (Cell Signaling Technology, Danvers, MA, USA, Cat 9661), CD44 (Thermo Fisher Scientific, Waltham, MA USA, Cat 156-3C11), ubiquitin (horseradish peroxidase (HRP) conjugate) (Cell Signaling Technology, Danvers, MA, USA, Cat 14049), anti-AKT (Cell Signaling Technology, Danvers, MA, USA, Cat 9272), anti-pAKT (Cell Signaling Technology, Danvers, MA, USA, Cat 9271S), anti-ERK(Abcam, Cambridge, UK, Cat ab184699), anti-pERK (Abcam, Cambridge, UK, Cat ab201015), anti-STAT3 (Cell Signaling Technology, Danvers, MA, USA, Cat 124H6), anti-pSTAT3 (Y705) (Cell Signaling Technology, Danvers, MA, USA, Cat 9145S), and β-actin (Abcam, Cambridge, UK, Cat ab8224). Anti-Flag (Cat F3165) and anti-HA (Cat H3663) antibodies from Sigma-Aldrich were used 1:1000 for Western blotting. The secondary antibodies that were used include anti-rabbit IgG HRP conjugate (Promega, Madison, WI, USA, Cat W401B) and anti-mouse IgG HRP conjugate (Promega, Madison, WI, USA, Cat W402B) for Western blotting, goat anti-mouse IgG (H + L) Alexa Fluor 405 (Thermo Fisher Scientific, Waltham, MA, USA, Cat A-31553), and goat anti-rabbit IgG (H + L) Alexa Fluor 568 (Thermo Fisher Scientific, Waltham, MA, USA, Cat A11011) for immunofluorescence, and mAb to IgG (HRP) (Abcam, Cambridge, UK, Cat ab131366) for immunoblotting after immunoprecipitation (IP). 

Plasmids that were used for overexpression include human CD44 standard form (NM_001001391), FLAG-tagged ORF Clone pCMV6-Flag-CD44s (OriGene Technologies, Rockville, MD, USA, Cat RC221820), CD44 full-length pCMV3-CD44f-HA ((Sino Biological, Beijing, China, Cat HG12211-CY), Lenti ORF clone of human MARCH8-GFP (OriGene Technologies, Rockville, MD, USA, Cat RC209891L2), GFP control (OriGene Technologies, Rockville, MD, USA, Cat PS10007), STAT3 (Addgene, Watertown, MA, USA, Cat 71450), and STAT3 Y705F (Addgene, Watertown, MA, USA, Cat 71445) mutant. 

### 2.6. Cell Apoptosis Assay

Detection of apoptosis of MDA-MB-231 and BT-549 cells was performed using the PE Annexin V Apoptosis Kit (BD Biosciences, Franklin Lakes, NJ, USA, Cat 559763) following the manufacturer’s instructions, with flow analysis on a BD LSR II flow cytometer (BD Biosciences, Franklin Lakes, NJ, USA, Cat 642221).

### 2.7. Colony Formation Assay

A total of 300 MDA-MB-231 cells were seeded onto 6 cm tissue culture plates containing 4 mL of medium and cultured continuously until macroscopic cell colonies were formed (up to 3 weeks). Then, the cells were fixed with 100% methanol and stained with 0.1% crystal violet solution for 10 min. After washes with PBS, the plates were photographed using a digital camera. Positive colony formation was calculated by manual counting. 

### 2.8. Western Blotting and Immunoprecipitation

Whole-cell lysates were made in RIPA lysis buffer (VWR, Radnor, PA, USA, Cat N653) containing Halt protease and a phosphatase inhibitor cocktail (Thermo Fisher Scientific, Waltham, MA, USA, Cat 78440). After quantification, equal amounts of protein of each sample were separated by SDS-PAGE and transferred to a nitrocellulose membrane. The membrane was blocked with 2% bovine serum albumin (BSA) in Tris-buffered saline containing 0.1% Tween-20 detergent (TBST) for 1 h at room temperature (RT), and then incubated with primary antibodies for 1 h at RT or 4 °C overnight, followed by three washes with TBST. HRP-conjugated secondary antibodies were used for a 1 h incubation at RT, followed by three washes with TBST. The membranes were developed with Pierce ECL2 Western blotting substrate (Thermo Fisher Scientific, Waltham, MA, USA, Cat 80196). The immunoblotting against β-actin was used as a loading control with the same membrane being reprobed after being blotted for other targets at distinct molecular weight or the lower part of the membrane being cut (away from the other target proteins) for the blotting. The full blots and density measurements are included in Appendix A, respectively. 

Total protein was isolated using Pierce™ IP Lysis Buffer (Thermo Fisher Scientific, Waltham, MA, USA, Cat 87787) with a protease inhibitor cocktail and incubated with antibodies and IgG for immunoprecipitation (IP) at 4 °C overnight. After antibody binding, Protein A/G PLUS Agarose Beads (Santa Cruz Biotechnology, Dallas, TX, USA, Cat sc-2003) were added and incubated at 4 °C for 4 h. Then, the protein-bound beads were washed and prepared for immunoblotting. Anti-FLAG conjugated beads (Sigma-Aldrich, Darmstadt, Germany, Cat M8823) and anti-HA-conjugated beads (Pierce Biotechnology, Rockford, IL, USA, Cat8836) were used in CD44 immunoprecipitation. 

### 2.9. Immunofluorescence Staining

Cells in a culture plate or on slides were fixed with 4% paraformaldehyde for 10 min, permeabilized with 0.25% Triton X-100 in PBS, and then blocked with 2% bovine serum albumin (BSA) in PBS for 1 h. All primary antibodies were incubated at 4 °C overnight. After three washes with PBS containing 0.1% Tween-20, cells were incubated with Alexa 405- or Alexa 568-conjugated secondary antibodies for 1 h, followed by three washes. Images were taken by a Nikon A1 MP Laser Scanning Confocal Microscope. 

### 2.10. Bioluminescence Imaging for Tumorigenesis and Lung Colonization

Mice were injected intraperitoneally (i.p.) with 100 μL of d-luciferin (30 mg/mL, GoldBio, St Louis, MO, USA). After 5–10 min, mice were anesthetized with isoflurane, and bioluminescence images were acquired using the IVIS Spectrum In Vivo Imaging System (Caliper Life Sciences, Waltham, MA, USA, Cat 124262). Signals are presented as total photon flux and were analyzed using Living Image 3.0 software. 

### 2.11. Lung Colonization Assay

For MDA-MB-231 cell-mediated colonization experiments, 1 × 10^5^ L2T-labeled MDA-MB-231 cells were injected into NSG mice via the tail vein. The lung colonization signal of metastatic tumor cells was monitored by bioluminescence imaging. At indicated times post injection, the lungs were removed for histology analysis. Mouse lung samples were fixed with 10% formalin and embedded with paraffin. Lung sections were stained with hematoxylin and eosin. The images were taken by a microscope equipped with an Olympus camera.

### 2.12. Cell Invasion in Wound Healing 

Plates were coated with 100 µg/mL Matrigel (Corning, NY, USA, Cat 354234) and incubated in a 37 °C CO_2_ incubator overnight prior to cell plating. Cells were plated in an image lock 96-well plate at a confluency of 25,000 cells per well. After 12 h, a scratch was created using the IncuCyte wound maker. After washing of the floating cells, the washed wells of adherent cells with the scratch wound were covered with another layer of Matrigel for 30 min at 37 °C in incubator. Then, culture medium was added to the plate. The filling of the scratch wound was monitored in real time by IncuCyte over 48 h. 

### 2.13. MPP-9 ELISA Assay

A total of 30,000 cells per well were seeded in a 24-well plate. After 48 h, cell culture supernatants were collected for measurement of MMP-9 antigen levels using a human MMP-9 ELISA kit (Sigma-Aldrich, Darmstadt, Germany,, Rab0372, Lot# 1207H173). The sandwich assay procedure was employed to detect MMP-9 levels with standard curves according to the manufacturer’s instructions.

### 2.14. Statistical Analysis

All quantitative experiments were performed with at least three independent biological replicates, and the results are presented as the mean ± SD. A Student *t*-test (two groups) was used to compare the mean of two groups of samples using GraphPad Prism 7 software, as shown in Figures. Human data analyses were obtained on an online platform with specific software, and *p* < 0.05 was considered statistically significant (* *p* < 0.05; ** *p* < 0.01; n.s., not statistically significant).

## 3. Results

### 3.1. Correlation between the Expression of MARCH8 and Breast Cancer and Breast Cancer Patient Overall Survival

To evaluate the association between MARCH family members and breast cancer in patients, we compared the mRNA levels of all MARCH genes in breast tumors with paired normal tissues in TCGA datasets using the online GEPIA platform [26]. The bioinformatic analysis showed that, among all MARCH family members, only MARCH8 and MARCH9 were significantly downregulated and upregulated, respectively, in breast tumors when compared with the normal tissue (Figure 1A). We further expanded the analysis of MARCH8 gene expression in all tumor types in TCGA. Among analyzed human cancers, MARCH8 was significantly downregulated in 10 additional tumor types when compared to the respective normal tissues, including bladder, cervical, colon, kidney, lung, rectum, thyroid, and uterine (Figure 1B); on the other hand, upregulated MARCH8 expression was observed in cholangial tumors, and no significant changes were found in stomach and other tumors (Figure 1B).

Moreover, using the Prognoscan analysis platform [27], we found that the expression levels of MARCH1, MARCH2, MARCH5, MARCH8, and MARCH10 were positively correlated with the overall survival of breast cancer patients, whereas higher expression levels of MARCH3, MARCH6, MARCH7, and MARCH9 had the opposite correlation with worse overall survival, as shown in Kaplan–Meier plots (Figure 2). Combining these two pieces of clinical relevance data, we hypothesized that MARCH8 is a tumor suppressor downregulated in breast cancer.

### 3.2. MARCH8 Expression Is Low in Breast Cancer Cells and Transient Restoration of MARCH8 Promotes Apoptosis of MDA-MB-231 and BT549 Cells

To test the hypothesis, we first analyzed the expression levels of MARCH8 in breast cancer cell lines. Using immunoblotting, we found that MARCH8 expression was low in breast cancer cells lines MCF-7, BT-549, SKBR3, BT-474, and MDA-MB-231 in comparison to the immortalized normal mammary epithelial cell line MCF-12A (Figure 3A). To investigate the function of MARCH8 in breast cancer cells, we restored MARCH8 expression by transient transfection of the cells with the vectors expressing the MARCH8-GFP fusion gene (Figure 3B). When compared to the cells transfected with a GFP control vector that caused minimal cell death, MARCH8-GFP-transfected cells showed a significant increase in annexin V^+^ apoptotic cells (60–80%) 2 days post transfection (Figure 3C,D). MARCH8 overexpression further decreased MMP-9 secretion and compromised cell invasion in wound healing on Matrigel-coated plates (horizontal invasion) (Figure 3E–G), both of which are related to metastasis. These data demonstrated that MARCH8 promotes apoptosis, and loss of MARCH8 provides advantages of cell survival and cell invasion for breast cancer cells.

### 3.3. Stable Expression of MARCH8 Inhibits Colony Formation In Vitro and Diminishes Tumorigenesis and Lung Colonization In Vivo

To determine the effect of MARCH8 overexpression on breast cancer development and progression in vivo, we generated stable MDA-MB-231 cell lines adapted to expression of MARCH8-GFP and control GFP after lentiviral transduction (Figure 4A,B). However, doxorubicin-mediated chemotherapeutic treatment resulted in proapoptotic signals in MARCH8-GFP-overexpressing cells, with higher levels of cleaved caspase-3 than in the GFP control cells (Figure 4B). Furthermore, MARCH8-GFP overexpression inhibited colony formation of breast cancer cells, demonstrating its role as a tumor suppressor in vitro (Figure 4C). 

To examine the functions of MARCH8 in tumor initiation and experimental metastasis (lung colonization) in vivo, we inoculated the L2T-labeled breast cancer cells into NSG mice orthotopically and intravenously, respectively, for bioluminescence imaging. Following orthotopic implantation of 100 cancer cells into the second and fourth mammary fat pads of NSG mice, bioluminescence imaging on day 14 revealed that GFP-control cells initiated tumor growth from three out of eight injections, whereas no tumor growth was observed in the mice implanted with MARCH8-GFP-overexpressing cells (Figure 4D). As MARCH8-GFP-overexpressing cells failed to initiation tumor growth following orthotopic implantation, no spontaneous metastasis was detected. Instead, we conducted an experimental metastasis assay to assess the lung colonization of these cells. Following tail-vein injection of 100,000 cells, MARCH8 expression compromised the lung colonization of breast cancer cells, with decreased bioluminescence signals and metastatic lesions in H&E-stained lungs analyzed on day 7 after injection (Figure 4E–G). The results support that MARCH8 inhibits tumorigenesis and lung colonization of MDA-MB-231 cells in vivo.

### 3.4. MARCH8 Mediates CD44 Degradation through the Lysosome Pathway

As reported by previous studies, CD44 is one of the glycoproteins downregulated by MARCH8 [28]. However, it was unclear whether CD44 has a direct interaction with MARCH8. CD44 is also a breast cancer stem-cell marker, contributing to cancer cell survival, circulating tumor cluster formation, and metastatic potential [20]. To assess whether CD44 is degraded by MARCH8 and/or contributes to the phenotype of MARCH8 in breast cancer cells, we performed immunoblotting, immunofluorescence (IF) staining, and flow cytometry analysis of CD44 in MARCH8-expressing cells. Overexpression of MARCH8 reduced CD44 protein levels with simultaneous upregulation of proapoptotic proteins BAX and BID in MDA-MB-231 cells (Figure 5A). Decreased CD44 levels in MARCH8-expressing cells were also confirmed by IF and flow cytometry compared to the GFP control cells (Figure 5B,C). Notably, the lysosome inhibitor chloroquine instead of the proteasome inhibitor MG-132 completely blocked MARCH8-mediated CD44 degradation (Figure 5D), suggesting that MARCH8 induces lysosome-dependent destruction of CD44. That is consistent with MARCH8 function in degrading other transmembrane glycoproteins [9,10,11,14]. 

To determine if CD44 is a possible substrate of MARCH8, with which it interacts, we overexpressed both MARCH8-GFP and CD44 (either standard form CD44s-FLAG or the full-length CD44f-HA) into HEK-293 cells for immunoprecipitation. In the presence of chloroquine, which inhibits lysosome-dependent protein degradation, both CD44 and MARCH8 were captured in the protein complex immunoprecipitated by anti-FLAG/HA-bound beads (Figure 5E), suggesting a possible interaction between MARCH8 and CD44 (CD44s and CD44f). To investigate the importance of loss of CD44 in MARCH8 phenotypic functions, we restored CD44 protein levels in MARCH8 cells via transient transfection of the CD44-FLAG cDNA plasmid. However, CD44 overexpression only slightly mitigated the levels of proapoptotic proteins BID and BAX, which were increased in MARCH8-GFP-overexpressing cells (Figure 5F), suggesting that additional targets of MARCH8 are responsible for the alteration of proapoptotic signals in breast cancer cells.

### 3.5. MARCH8 Mediates STAT3 Degradation through the Proteosome Pathway

In the search for other possible targets of MARCH8 regulating cell survival, we analyzed the total protein levels and activation by phosphorylation of multiple signaling transducers known to regulate cell survival, such as AKT, ERK, and STAT3, in MDA-MB-231 breast cancer cells that stably express MARCH8-GFP and control GFP. Cells were cultured as both adherent and in suspension to mimic the detached status of metastatic cancer cells. MARCH8 did not alter the total protein levels or phosphorylation status of AKT and ERK in either adherent or suspension cultures (Figure 6A). Although the total protein levels of STAT3 remained stable in MARCH8-expressing and GFP control cells in both culture conditions, the phosphorylated STAT3 at Y705 was remarkably downregulated by MARCH8 in cells in suspension and less dramatically altered in adherent cells (Figure 6B). The decreased pSTAT3 levels were completely restored by the proteasome inhibitor MG-132 but not by the lysosome inhibitor chloroquine (Figure 6B). In contrast, MARCH8-mediated CD44 degradation was partially blocked by chloroquine but not by MG-132 (Figure 6B).

To determine whether STAT3 is a ubiquitination substrate of MARCH8, we assessed their interactions and possible ubiquitination of STAT3. After co-transfection of STAT3 (or Y705F mutant) and MARCH8-GFP (or GFP vector control) and a short treatment with MG-132 to inhibit proteasomal degradation, HEK-293 cell lysates were immunoprecipitated by anti-STAT3 or anti-pSTAT3 Y705. Indeed, MARCH8 was found in the protein complex with both the wildtype STAT3 and the Y705F mutant with compromised phosphorylation (Figure 6C). Notably, pSTAT3 at Y705 (pulled down by anti-pSTAT3) had minimal interaction with MARCH8 and undetectable ubiquitination, whereas the mutated STAT3^Y705F^, which was marginally precipitated by the anti-pSTAT3, had much more abundant MARCH8 protein in the complex, coupled with extraordinary levels of ubiquitination (Figure 6C), implicating potential competition between STAT3 phosphorylation at Y705 and ubiquitination.

Lastly, we investigated if STAT3 is an essential target of MARCH8 by transient transfection of wildtype or mutant STAT3^Y705F^ into MARCH8-GFP-expressing MDA-MB-231 breast cancer cells. Compared to the GFP control cells, MARCH8 expression upregulated proapoptotic proteins BID and BAX, which were inhibited neither by CD44 restoration nor by wild-type STAT3 overexpression, but almost completely rescued by STAT3^Y705F^ overexpression, with undetected BID and BAX in suspension cells in particular (Figure 6D). The accumulation of STAT3^Y705F^ suggests possible resistance of this mutant to degradation. 

Taken together, our results demonstrate that STAT3 and CD44 are newly discovered targets of MARCH8 in breast cancer which can regulate cell survival and metastasis.

## 4. Discussion

Our studies reveal that tumor suppressor MARCH8 is specifically downregulated in breast cancer and additional cancer types, such as bladder, cervical, colon, kidney, lung, rectum, thyroid, and uterine. Consistently, overexpression of MARCH8 has been shown to inhibit non-small-cell lung cancer (NSCLC) cell proliferation and metastasis via the phosphoinositide 3-kinase and mTOR signaling pathways and induced apoptosis of A549 and H1299 cells [29], demonstrating the antitumor properties of MARCH8 in NSCLC. MARCH8 was also reported to downregulate TRAIL-R1 cell surface expression in MCF-7 breast cancer cells [30], suggesting various substrates to be identified in a cell context-dependent manner. However, the role of MARCH8 in each of the other tumor types requires further investigation. In gastric and esophageal cancers, MARCH8 expression was not downregulated in comparison to normal adjacent tissues, and its functions might be context-dependent [31,32]. 

MARCH8 and many other MARCH E3 ligases are known to target various membrane proteins for protein trafficking and degradation, as well as regulation of transcription and DNA repair [9,11]. Our study reports a novel function of MARCH8 in ubiquitinating and degrading a nonmembrane protein, STAT3, through the proteasomal degradation pathway, in addition to a more typical membrane protein target, CD44, through the lysosomal degradation pathway. MARCH8 is expressed widely in early and late endosomes of human tissues and cells, and it plays an important role in immune responses [29,31,32,33,34]. In this study, we elucidated the function of MARCH8 in cell survival, tumorigenesis, and metastasis in breast cancer, in which CD44 [15,20,21,35] and STAT3 [22,23] are known to contribute to the phenotypic functions mediated by cancer stem cells. 

Future studies will further identify the biochemical properties of MARCH8 in mediating the interactions with CD44, STAT3, and other new targets, map their ubiquitination sites in connection to lysosome and proteasome degradation pathways, and elucidate the interplay with phosphorylation-mediated alterations of cell fate, stem-cell functions, and metastasis in breast cancer, as well as other cancers.

## 5. Conclusions

Our study identifies MARCH8 as a new tumor suppressor in inhibiting breast cancer metastasis and enhancing cancer cell death, partially via the lysosomal degradation of the breast cancer stem-cell marker CD44, as well as via the proteosome-dependent degradation of STAT3. 

## Figures and Tables

**Figure 1 cancers-13-02550-f001:**
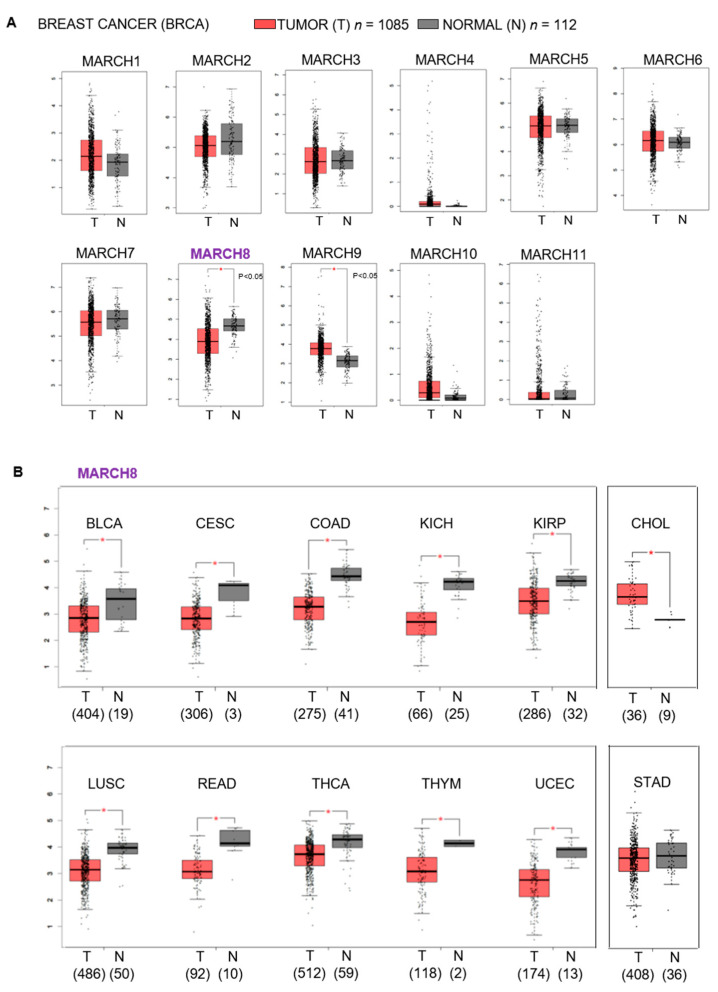
MARCH family gene expression levels in human tumors and normal tissues in TCGA. (**A**) Boxplots of mRNA expression levels of MARCH family genes (*MARCH1*–*MARCH11*) between human breast tumors (T, red box, *n* = 1085) and paired normal tissues (N, gray box, *n* = 112) in TCGA dataset via the online GEPIA platform accessed on 19 March 2021 (http://gepia.cancer-pku.cn/). MARCH8 expression is downregulated and MARCH9 expression is upregulated in breast tumors versus paired normal tissues (* *p* < 0.05). (**B**) Boxplots of MARCH8 mRNA expression levels of (Ensembl ID: NSG00000165406.15) in multiple human tumors (T, red box) versus respective normal tissues (N, gray box) in TCGA dataset, with downregulation in 10 tumors, including BLCA (bladder urothelial carcinoma), CESC (cervical squamous cell carcinoma and endocervical adenocarcinoma), COAD (colon adenocarcinoma), KICH (kidney chromophobe), KIRP (kidney renal papillary cell carcinoma), LUSC (lung squamous cell carcinoma), READ (rectum adenocarcinoma), THCA (thyroid carcinoma), THYM (thymoma), and UCEC (uterine corpus endometrial carcinoma). Upregulated expression of MARCH8 observed in CHOL (cholangio carcinoma) and no significant difference in STAD (stomach adenocarcinoma) compared to respective normal tissues. The sample size of the tissues (n) included in the parentheses below T or N. The *Y*-axis of gene expression is log_2_ (transcripts per million + 1) on a log scale. Asterisks (*) represent significant differential expression between tumor (T) and paired normal (N) tissues in TCGA, with cutoffs of log_2_ (fold change) = 0.5 and *p* < 0.05. A jitter size of 0.4 was used in the boxplots.

**Figure 2 cancers-13-02550-f002:**
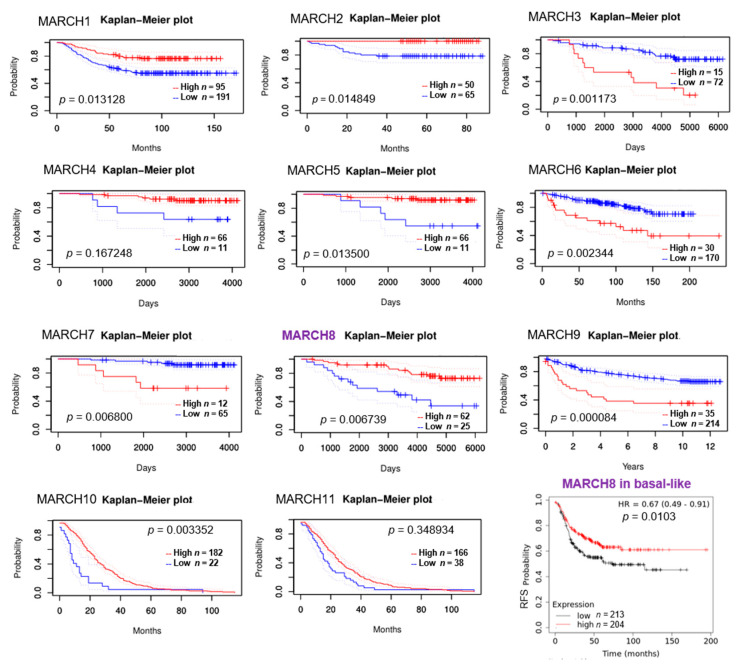
Association of MARCH8 expression and other family members with overall survival in breast cancer patients. A higher expression of MARCH8 and other MARCH family members (MARCH1, 2, 5, 10) is correlated with a higher overall survival probability for breast cancer patients according to the Prognoscan analysis. MARCH8 mRNA expression levels are also associated with relapse-free survival (RFS) of basal-like breast cancer (log rank *p* = 0.01).

**Figure 3 cancers-13-02550-f003:**
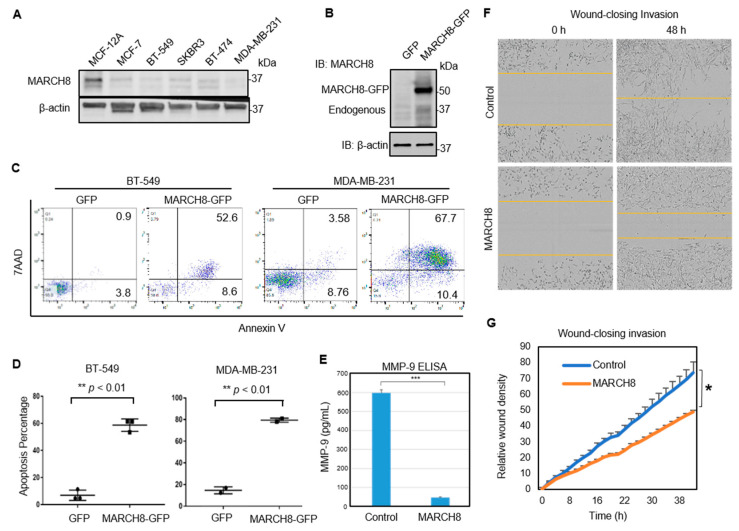
MARCH8 overexpression induces cell apoptosis, reduces MMP-9 levels, and compromises cell invasion. (**A**) Immunoblots of endogenous MARCH8 protein expression (~37 kDa) in immortalized normal mammary epithelial cell line MCF-12A and breast cancer cell lines including MCF-7, BT-549, SKBR3, BT-474, and MDA-MB-231. (**B**) Immunoblots of MARCH8 indicating the overexpression of fused MARCH8-GFP (~50 kDa) in MDA-MB-231 cells after transient transfection in comparison to a GFP vector control. (**C**,**D**) Transient transfection of MARCH8 significantly promotes cell death in BT-549 and MDA-MB-231 breast cancer cells, as indicated by annexin V levels (apoptosis) and 7AAD (DNA dye). ** *t*-test *p* < 0.01. (**E**) ELISA-detected MMP-9 levels in the supernatant of the control and MARCH8-overexpressing cells. *** *t*-test *p* < 0.001. (**F**,**G**) Images (**F**) and cell invasion-based wound closure curves (**G**) of MDA-MB-231 control and MARCH8-overexpressing cells between 0 and 48 h after wound scratch, analyzed by Incucyte time-lapse imaging. * *p* < 0.05.

**Figure 4 cancers-13-02550-f004:**
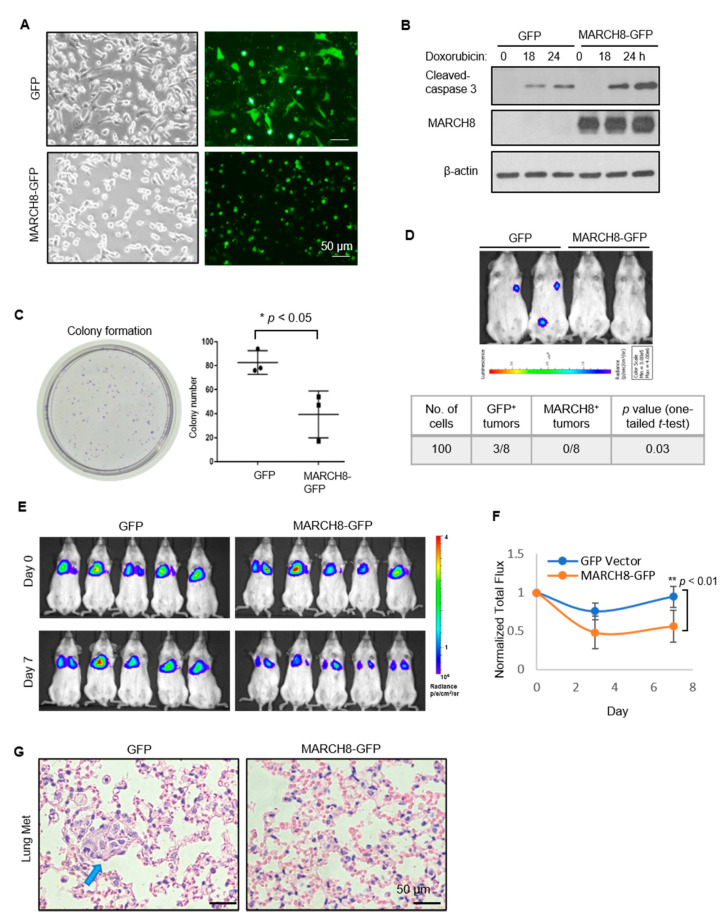
MARCH8 inhibits colony formation in vitro and lung colonization in vivo. (**A**) Cell images with stable expression of MARCH8-GFP and control GFP after lentiviral transduction of MDA-MB-231 cells. (**B**) Immunoblots of proapoptotic protein, cleaved caspase 3, and MARCH8-GFP indicating an increase in 10 µM doxorubicin-induced apoptosis in the MARCH8-GFP-overexpressing cells. (**C**) Colony formation image (left panel) and bar graph (right panel) of compromised colony formation in MDA-MB-231 cells with stable expression of MARCH8-GFP in comparison to the control GFP cells in vitro. * *t*-test *p* < 0.05 (**D**) Bioluminescence images (top panel) and a summary table (bottom panel) of in vivo tumorigenesis showing that three out of eight orthotopic inoculations with 100 GFP control cells into mouse mammary fat pads formed possible tumors 14 days after tumor cell implantation, whereas MARCH8-GFP-overexpressing cells formed no tumors out of eight inoculations (0/8). (**E**,**F**) Bioluminescent images (**E**) and quantifications (**F**) of the lung colonization mediated by the GFP control cells with higher metastasis signals and MARCH8-GFP-overexpressing cells with lower metastasis signals 7 days after tail vein injection of the tumor cells. ** *t*-test *p* < 0.01 (**G**) H&E staining of the lungs removed from the mice 7 days post tail-vein injection of GFP and MARCH8-GFP-expressing cells. The blue arrow points to the metastatic tumor cells disseminated into the lungs.

**Figure 5 cancers-13-02550-f005:**
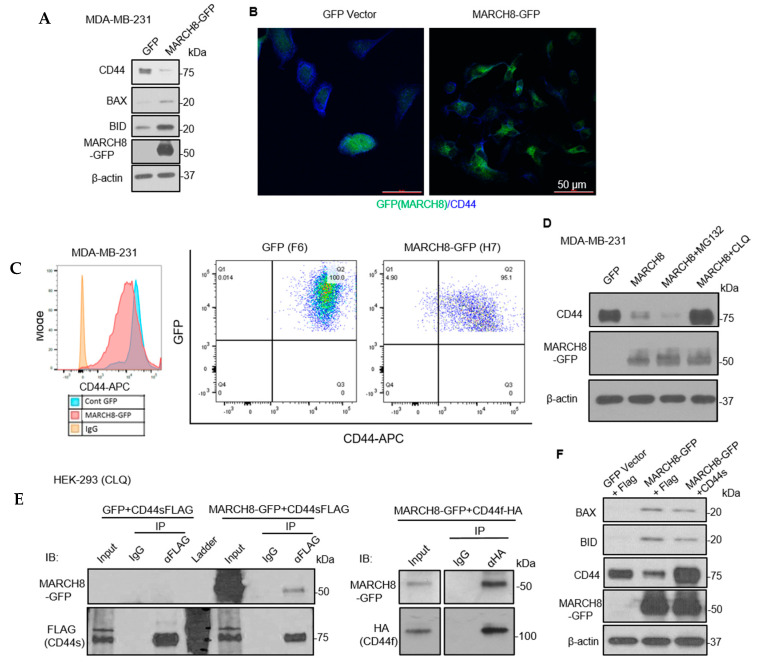
MARCH8 interacts with and degrades CD44 through the lysosome pathway. (**A**) Immunoblots to detect decreased CD44 and increased BAX, BID, and MARCH8 expression levels in MARCH8-overexpressing cells compared to GFP control cells. (**B**) Immunofluorescence staining with anti-CD44 antibody showing decreased expression of membrane protein CD44 (blue) in MARCH8-GFP expressing cells. (**C**) Flow cytometry histogram overlay (left panel) and dot plots (right panels) indicating MARCH8-decreased CD44 expression levels in negative association with MARCH8-GFP signals. (**D**) Immunoblots of endogenous CD44 and exogenous MARCH8-GFP after transient transfection of MARCH-GFP and treatment with MG-132 or chloroquine (CLQ) to block the proteasomal or lysosomal degradation pathways, respectively. (**E**) Immunoblots of MARCH8 and FLAG-tagged CD44 after anti-FLAG mediated immunoprecipitation (IP) of the lysates of HEK-293 cells after transfections with MARCH8-GFP and CD44-FLAG (standard isoform CD44s and full-length CD44f) and treatment with CLQ, indicating the interactions between MARCH8 and CD44 (CD44s or CD44f). (**F**) Immunoblots of BAX, BID, CD44, and MARCH8 in the MDA-MB-231 cells with stable expression of GFP or MARCH8-GFP with transient transfection of a FLAG vector control or restoration of CD44 expression via CD44s-FLAG. CD44 overexpression slightly inhibited the expression of proapoptotic BID and BAX in MARCH8-GFP-overexpressing cells.

**Figure 6 cancers-13-02550-f006:**
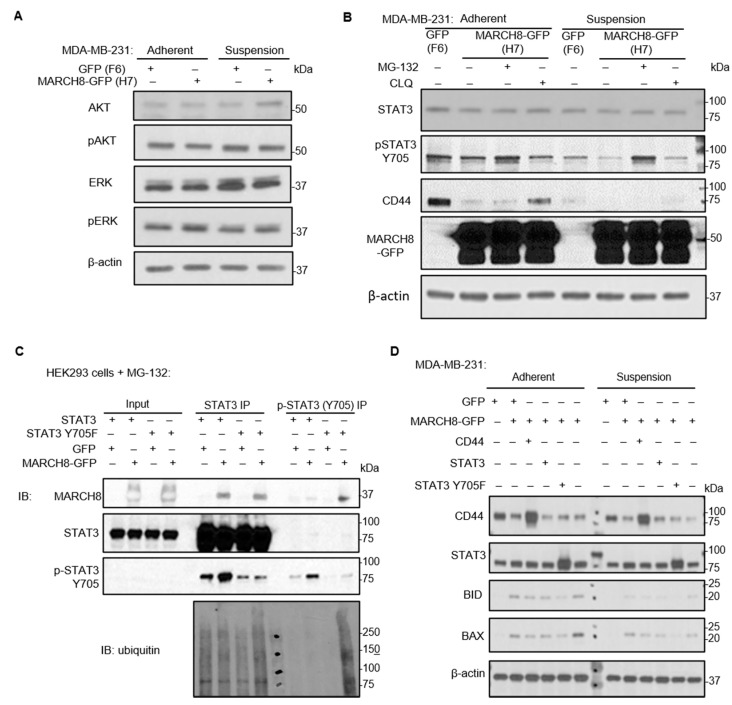
MARCH8 interacts with and degrades STAT3 through the proteasome pathway. (**A**) Immunoblots of AKT, pAKT, ERK, and pERK in MDA-MB-231 cells with stable expression of GFP and MARCH8-GFP, both adherent and in suspension. (**B**) Immunoblots of STAT3, pSTAT3 (Y705), CD44, and MARCH8 in GFP- and MARCH8-GFP-expressing cells in the absence or presence of MG-132 and CLQ, both adherent and in suspension. (**C**) Immunoblots of MARCH8, STAT3, pSTAT3 (Y705), and ubiquitin in lysates of cells co-transfected with STAT3 (wildtype or Y705F) and MARCH8-GFP (or GFP control) and immunoprecipitated by anti-STAT3 and anti-pSTAT3 (Y705). (**D**) Immunoblots of CD44, STAT3, BID, and BAX in MDA-MB-231 cells with stable expression of GFP and MARCH8-GFP, with transient transfection with CD44, STAT3, or mutant Y705F as indicated, both adherent and in suspension.

## Data Availability

Data will be shared with the community upon request to corresponding authors.

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
