# Peer review of "MARCH8 Suppresses Tumor Metastasis and Mediates Degradation of STAT3 and CD44 in Breast Cancer Cells"

_cancers, 2021, doi:10.3390/cancers13112550_

Round 1
Reviewer 1 Report
This work focused on studying the role of a RING E3 ligase - MARCH8, which is member of the membrane-associated RING-CH (MARCH) family. It provided experimental evidence indicating that MARCH8 acted as a tumor suppressor by inhibiting breast cancer metastasis and enhancing cancer cell death. Mechanistically, MARCH8 promoted lysosomal degradation of the breast cancer stem cell marker CD44, and simultaneously it induced STAT3 degradation via the proteosome-dependent pathway. Data presented are clear and convincing. The information included should be very interesting to the investigators on identifying novel functions of E3 ligases in breast cancer.
Specific comments:
- It seemed that database analyses of MARCH8 in Figs. 1 & 2 were performed using all breast cancers. It was not clear whether a similar trend would be obtained in the subtype of triple negative breast cancer.
- There was no description of figure 4G in the text. It lacked figure legends for Fig. 4G.
Author Response
Dear Reviewer 1, thanks for great comments and we appreciate your time and effort. Please see attached response letter with point-by-point answers to your comments followed by both clean and tracked versions of rthe evised manuscript. Have a nice day!
-Huiping

Reviewer 2 Report
In this study, the authors proposed an interesting role of MARCH8 in regulating breast cancer development through degradation of STAT3 and CD44. The studies partially support this proposal, but some methodological and technical issues limit the reliability of the conclusions. There are some issues that need clarification before this manuscript could be recommended for publication.
General comments:
1). My main doubts relate to the statement that MARCH8 regulate TNBC only. The patient data (Figure 1 and Figure 2) as I understand it apply to general breast cancer cases and not just TNBCs. Moreover, the authors showed in Figure 3A that a decrease in MARCH8 protein levels is also observed in MCF-7, which are non-TNBC cells. It therefore seems that a reduction in MARCH8 levels is an inherent feature of breast cancer cells, and not only related to TNBC.
2). Secondly, the authors' statement that MARCH8 suppresses tumor metastasis is not covered by the experiments performed. The Authors performed Lung Colonization Assay which assess the capacity of cancer cells to seed, survive, and formation of colonies in the lungs after intravenous injection rather than metastasis. Please refer to: Cancer Res. 2006 Apr 1;66(7):3386-91. doi: 10.1158/0008-5472.CAN-05-4411 and Dis Model Mech. 2017 Sep 1;10(9):1061-1074. doi: 10.1242/dmm.030403. Due to the fact that animal studies are costly and time-intensive, alternative biochemical and functional tests can be performed on cell models. In order to confirm the hypothesis, I suggest to perform the classical test of cell transmigration through the matrigel or through the endothelial cell monolayer and additionally determine the secretion of metalloproteinases, which are known to play a key role in modulating cell metastasis.
3). The discussion should be rewritten and must focus more on the results obtained by the authors in the context of already known results in the field.
Specific comments:
4). More detailed information on the methodology should be given. E.g., I couldn’t find any detailed information about the procedure for stripping and re-probing a membrane. Did the authors use the compensation method in the FACS analysis? What was the average quality of the RNA extracted (A230/280 ratio)? Why was only one housekeeping gene selected? Has any analysis been performed based on any software (for example geNorm or any others)? Please provide the sequence of the primers used.
5). Figure 4A – please provide a better quality of images. Figure 4B – please provide representative images for GFP and f MARCH8-GFP MDA-MB-231 cells; Figure 4G - scale bars are missed.
Author Response
Dear Reviewer 2,
We are grateful to you for your insightful suggestions and instructive comments. Attached please see the response letter with point-by-point answers to address all of your comments followed by both clean and tracked versions of the revised manuscript. We appreciate your time and effort. Have a nice day!
-Huiping Liu

Round 2
Reviewer 2 Report
I appreciate the authors put much effort to fix the issues that were mentioned in the original review. The corrections implemented by the Authors improve the quality of the manuscript, however there are still some weakness that should be fixed before manuscript could be recommended for publication. Please find specific comments below:
1). The most important issue, not corrected by the Authors, is the statement that "Stable Expression of MARCH8 Inhibits ... Metastasis In Vivo". Since Lung Colonization Assay does not define cell invasion or metastasis, the authors' statement is incorrect and needs to be corrected. Major redrafting of the description of animal studies and deletion of metastasis information should be undertaken. The information provided is misleading to readers and is not supported by the experimental data.
2). As the Authors performed non typical invasion assay – wound healing assay in the matrix environmental instead of the classical test of cell transmigration through the membrane with pore covered with Matrigel, a detailed description within results section should be included. Otherwise, readers may be under the illusion that the results are for wound healing assay and not for an invasiveness assay.
3). The Authors stated, that they used separate gels and membranes for proteins of interest and beta-actin as a loading control. Many WB specialists consider such an analysis of the protein level to be an essential error because it prevents proper analysis. The loading control should be analyzed on the same gel as it determines the potential variation in loading the sample onto the well. If the authors analyzed the loading control on another gel, it should be clearly stated both in the description of the method and the figure's description. Readers must be advised that the results are comprised of independent gels and membranes.
Author Response
Thanks for additional comments. We addressed all in the attached letter.
